# Understanding the trends, clustering, and risk factors of pinworm infection in preschool settings: A repeated cross-sectional multi-center study between 2019 and 2024

Fanzhen Mao[1,2☉], Xuecheng Li[1☉], Sheng Ye[1], Feng Tang[2], Bixian Ni[2], Qiang Zhang[2], Jiayao Zhang[2], Yaobao Liu[1,2*], You Li[1,3,4*], Jun Cao[1,2*]

**1** School of Public Health, Key Laboratory of Public Health Safety and Emergency Prevention and Control Technology of Higher Education Institutions in Jiangsu Province, Nanjing Medical University, Nanjing, China, **2** Jiangsu Institute of Parasitic Diseases, National Health Commission Key Laboratory of Parasitic Disease Control and Prevention, Jiangsu Provincial Key Laboratory on Parasite and Vector Control Technology, Wuxi, China, **3** Centre for Global Health, Usher Institute, University of Edinburgh, Edinburgh, United Kingdom, **4** Changzhou Third People's Hospital, Changzhou Medical Centre, Nanjing Medical University, Changzhou, China

☉ These authors contributed equally to this work.
* yaobao0721@163.com (YBL); you.li@njmu.edu.cn (YL); caojuncn@hotmail.com (JC)

## Abstract

### Background

Pinworm (*Enterobius vermicularis*) remains the most prevalent helminth among preschool-aged children worldwide. Despite China's rapid development and decades of mass deworming, pinworm infection persists as a significant paediatric public-health problem. In this study, we aimed to understand the trends, clustering, and risk factors of pinworm infection in preschool settings of Jiangsu, China.

### Methods

A repeated cross-sectional study was conducted in 45 counties of Jiangsu Province from 2019 to 2024. Multi-stage cluster convenience sampling was applied. Each year, one rural preschool and one urban preschool were selected per county. The sample size for each school was 229 to obtain a 5% margin of error, an arbitrary design effect of 1.5 for a prevalence estimate of 10%, and a 10% nonresponse rate. A total of 27,925 children were investigated. Adhesive cellophane tape swabs were collected for confirmation of pinworm infection. A preschool-level permutation approach was used to evaluate clustering effects of pinworm infections within preschools, measured by infection rate ratio (IRR). Risk factors of pinworm infections were assessed by multiple logistic regression. Mediation analysis was conducted between the risk factor and pinworm infection.

**Data availability statement:** The datasets generated and/or analysed during the current study are available in the Zenodo repository via https://doi.org/10.5281/zenodo.17119450.

**Funding:** This work was supported by the Jiangsu Commission of Health Medical Research Project (M2022064 to FM), the Jiangsu Province Capability Improvement Project through Science, Technology, and Education (ZDXYS202207 to JC), and Jiangsu Provincial Health International (Regional) Exchange Support Program (2023 Project to FM). The funders had no role in the conceptualization, design, data collection, analysis, decision to publish, or preparation of the manuscript.

**Competing interests:** The authors have declared that no competing interest exists.

## Results

The overall pinworm infection rate was 0.48%, ranging form 1.22% in 2019 to 0.11% in 2024 and showing a downward trend ($\chi^2$=52.436, $P$<0.001). Significant within-preschool clustering effect was observed in 2019 (IRR=5.95, 95% lower CL=3.77) and 2021 (IRR=1.96, 95% lower CL=1.35). Risk factors for preschool pinworm infection included migrant children (OR=3.911, 95%CI=2.749-5.610) and older age (OR=1.443, 95%CI=1.209-1.730). Mediation analysis indicated that parental education and family income collectively explain 55.2% of the association between migrant status and infection.

## Conclusions

Pinworm infections among preschool children demonstrate a within-preschool clustering effect. Despite the observed decline in infection rate of pinworms, targeted interventions are necessary in preschools, especially for migrant children. This study contributes to the broader understanding of enterobiasis and support the development of strategies to protect children's health in preschools.

### Author summary

Pinworm infection is among the most prevalent helminth infections in young children worldwide, yet we still lack long-term, multi-center data on how it spreads and whether infections cluster within preschools. Our study aims to chart the trend of pinworm infection rates, quantify the within-preschool clustering effect, and identify risk factors contributing to the pinworm infections in preschool settings. From 2019 to 2024 we sampled nearly 28,000 children across 104 preschools in Jiangsu, China, using a standardized swab-and-microscope method. A preschool-level permutation approach was used to evaluate clustering effects of pinworm infections within preschools, measured by infection rate ratio (IRR). Over six years the overall infection rate fell. Pinworm infections among preschool children demonstrate a within-preschool clustering effect in 2019 and 2021, confirming that transmission clusters within individual schools rather than randomly across communities. Migrant children and older age were significant risk factors. Mediation analysis indicated that parental education and family income collectively explain 55.2% of the association between migrant status and infection. These findings show that even in a low-prevalence region, pinworm can hide in socially disadvantaged pockets; targeting these specific populations with hygiene upgrades and periodic screening could accelerate local elimination and serve as a model for other regions striving to eliminate neglected helminth infections.

## Introduction

Pinworm infection, caused by *Enterobius vermicularis*, is among the most prevalent helminth infections in young children worldwide [1,2], irrespective of socioeconomic

status, cultural background, or geographical location [3,4], occurring at a prevalence of 4–28% in children [5,6]. Despite advancements in public health initiatives and improvements in living standards, pinworm infections remain a significant public health concern due to their potential impact on children's physical and cognitive development [7–10]. The infections are particularly concerning in preschool settings [11], where children's close interactions can facilitate the spread of the parasite, leading to outbreaks and highlighting the need for targeted interventions.

China has achieved remarkable success in controlling parasitic diseases, leading to a substantial decline in population-level infection rates. Nevertheless, these diseases have not been eliminated, and pinworm infections in children remain common. Large-scale population surveys are particularly time-consuming and resource-intensive [12], posing a major challenge for investigating pinworm infections in preschool children. Consequently, despite the enduring prevalence of pinworm infections, particularly among preschool-aged children, the trend of these infections within preschool settings remains obscure. Moreover, it is widely assumed that pinworm infection clusters within preschools and that children attending preschool have higher infection rates than children living at home, but this assumption has never been formally examined. Frequent close contacts and still-developing hygiene habits make preschool children particularly susceptible to pinworm infections, resulting in markedly higher prevalence rates. The phenomenon of infection clustering within preschool settings has yet to be conclusively established, leaving a gap in our understanding of how pinworm infections propagate in these close-contact environments.

Thus, this study aims to chart the trend of pinworm infection rates, quantify the within-preschool clustering effect, and identify risk factors contributing to the pinworm infections. By analyzing data from a six-year multi-center study, this study endeavors to shed light on the epidemiological aspects of enterobiasis and inform the development of targeted interventions to protect children's health.

## Methods

### Ethics statement

This study was approved by the Institutional Review Board of Jiangsu Institute of Parasitic Diseases (JIPD-2018–002), Wuxi, China. Written informed consent was obtained from the parent or legal guardian of each child participant prior to enrollment.

### Study design, setting, and participants

A six-year multi-center cross-sectional study was conducted across Jiangsu Province, China from 2019 to 2024. Situated in the lower reaches of the Yangtze River in Eastern China, Jiangsu Province features a climate with favorable temperature and humidity conditions that support the transmission of intestinal parasites. Multi-stage cluster convenience sampling was applied [13]. Simple random sampling required a sample size of 208 to obtain a 5% margin of error, and an arbitrary design effect of 1.5 for a prevalence estimate of 10%, as reported in a previous study [14]. The final sample size for each preschool was 229 individuals, including a 10% nonresponse rate. A rural preschool and an urban preschool were selected from each county [15]. Totally 45 counties and 104 preschools were included in the study (Fig 1). Study participants were 27,925 preschool children aged 2–7 and their parents or caretakers. Classes from each grade (lower, middle, and upper grade) were randomly chosen in a preschool to balance the age bracket and meet the sample size. All students in the selected class were eligible and were enrolled after providing signed informed consent.

### Data collection

The surveys were conducted once per year, always during one of the two fixed windows: either May-June or from September-October. Standardized questionnaires including socio-demographic characteristics and hygiene habits, were distributed to parents or primary caregivers, facilitated by teachers. Children delivered these questionnaires to their

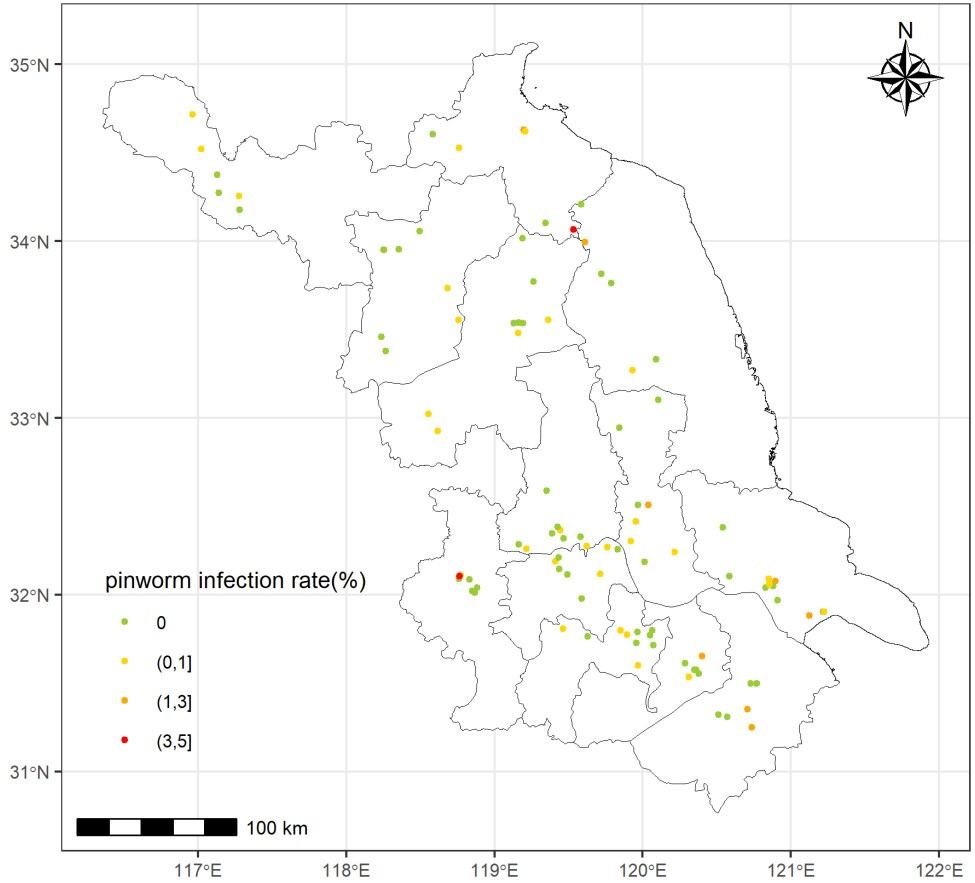

**Fig 1. Location and pinworm infection rate of preschools in the study.** Administrative boundaries from GADM (https://gadm.org), CC BY 4.0.

parents and returned the completed forms to their teachers. The questionnaire simultaneously recorded whether children had experienced symptoms associated with pinworm infections (stomachache, diarrhea, fever, painful urination, and perianal pruritus) within the past month. To minimise systematic and random bias, the following quality-assurance procedures were implemented: (1) The questionnaire was standardized and validated in a pilot study [13]. (2) Data collectors were uniformly trained before investigations. (3) Completed questionnaires were reviewed onsite for completeness and logic, and data were double-entered independently into EpiInfo 3.5.4; discrepancies were resolved against the originals until 100% concordance was reached. Sample collection took place in the morning at the preschool, prior to the children's bowel movements and baths. Health workers from the local Center for Disease Control and Prevention collected one sample per child. The adhesive cellophane tape swab (Patent Number: ZL201420707045.8) was applied to the perianal skin and subsequently examined by a trained microbiologist. All positive and 10% of negative samples underwent re-check. Any discrepancies in the results were resolved by a panel of experts.

## Variables and measurement

The primary outcome was pinworm infection, defined as the detection of *Enterobius vermicularis* eggs of the cellophane tape swab sample on the perianal skin. Infection status was considered positive if the swab contained eggs. Age, gender, parents' education level, migrant status, family income level, and children's hygiene habits were extracted from the

standardized questionnaires completed by parents or primary caregivers. Family income level was classified according to the OECD-modified tercile approach commonly used in Chinese health-equity research (NDRC Macro-team, 2025). Equivalised annual disposable income was calculated by dividing the total household income by the square root of household size. Low income was defined as < 50% of the national median, medium income as 50–200% of the national median, and high income as > 200% of the national median. Based on the 2025 China Household Income Survey, these cut-offs correspond to approximately < ¥90 000, ¥90 000–¥360 000 and > ¥360 000 per year for urban households, and < ¥36 000, ¥36 000–¥144 000 and > ¥144 000 for rural households (CPI-adjusted to the survey year) [16]. Hand-washing before meals, hand-washing after toilets, keeping fingers short, sucking fingers, sucking toys/pens, drinking unboiled tap water or not, and having separate towl were recorded as binary variables (0 = no, 1 = yes) based on parental report. Bathing frequency was captured as ordinal variable reflecting the number of episodes per week (1 = everyday, 2 = 2–3 times per week, 3 = ≤once per week). Bed-linen drying-out frequency was also captured as ordinal variable (1 = once per month, 2 = once per season, 3 = ≤once every six months). Symptoms experienced during the past month (e.g., stomachache, fever, perianal pruritus) were dichotomised into binary variables (0 = absent, 1 = present).

## Analyses

Descriptive statistics were employed to characterize the participants and potential influencing factors, while the Chi-square test was utilized to assess significant differences between group outcomes. The Chi-square test for trend was employed to investigate the trend in infection rates across different years.

A spatial autocorrelation analysis using Moran's I was conducted in ArcGIS Pro to assess the clustering patterns of infection rates across different regions. A permutation approach was applied to assess the clustering effect of pinworm infections in preschools; as null hypothesis, the approach assumed that the variations in pinworm infections across preschools were due to chance. Intuitively, if infections are truly random, every child should have the same chance of being infected regardless of which preschool they attend; in that case, any apparent "hot-spots" would simply reflect ordinary sampling variation. By contrast, if some schools harbour hidden risk factors (e.g., poor hygiene practices, shared bedding, or asymptomatic carriers), we should observe unusually high infection rates that cannot be explained by chance alone. We quantified the such variations by grouping preschools into high-risk and low-risk groups using the expected infection rate as cut-off, and calculating the Infection Rate Ratio (IRR) between high-risk and low-risk groups (as observed IRR). Then we generated 1,000 null datasets by randomly reassigning infection status while keeping the total number of cases constant, each time recalculating the IRR (as permuted IRR). Subsequently, we calculated the ratio of IRR between IRR from the observed data and the permutated data (1,000 sets) as the adjusted IRR, as shows in the formula:

$$\text{IRR} = \frac{\text{Observed high} - \text{risk group rate}/\text{Observed low} - \text{risk group rate}}{\text{Permuted high} - \text{risk group rate}/\text{Permuted low} - \text{risk group rate}} \tag{1}$$

An adjusted IRR > 1 indicates excess infection risks within a preschool (one-sided test), while an IRR close to 1 suggests random variations in infection risks between preschools. We further conducted a power analysis by simulating an effect of IRR = 2.0 to aid the interpretation of non-significant findings.

Multiple stepwise Logistic regression was applied to analyze the risk factors for pinworm infection. To examine the mechanisms underlying the association between migrant status and pinworm infection, we conducted a structural equation model (SEM) with parental education and family income level as parallel mediators. Average education level of both parents and family income level were entered as mediators; infection status was treated as binary. The model was fitted with the WLSMV estimator using the lavaan package. Natural indirect (NIE) and natural direct (NDE) effects were defined and 95% CIs were obtained with the Delta method. The proportion mediated (NIE/total effect) quantified the magnitude of mediation. We conducted all analyses in R version 4.3.1 within the Rstudio IDE (version 2023.09.0). The significance level was set at 0.05.

## Results

The spatial pattern is likely to be a result of random chance, as indicated by the z-score = 1.906 ($P$ = 0.057), according to the spatial autocorrelation (Moran's I) report (S1 Fig). A total of 27,925 preschool children were involved, comprising 14,519 (52.0%) boys, with an average age of 4.6 years. Descriptive statistics are presented in Table 1.

### Notable decrease in pinworm infection rate from 2019 to 2024

A total of 134 children were tested positive for pinworm infection. The overall pinworm infection rate over the six-year period from 2019 to 2024 was 0.48%. The annual infection rates were as follows: 1.22% in 2019, 0.19% in 2020, 0.49% in 2021, 0.28% in 2022, 0.11% in 2023, and 0.11% in 2024, indicating a general downward trend ($\chi^2$ = 52.436, $P$ < 0.001). The overall pinworm infection rate for rural preschools over the six-year period was 0.55%. A significant decline in pinworm infection rates was observed in rural preschools across the years ($\chi^2$ = 6.978, $P$ = 0.008). The overall pinworm infection rate for urban preschools was 0.44%. Significant reductions were noted in urban preschools over the years ($\chi^2$ = 58.462, $P$ < 0.001). The pinworm infection rates in preschools are show in Fig 2 (by year) and Fig 3 (by city).

### Within-preschool clustering of pinworm infections

Table 2 presents IRR for pinworm infection among preschool children in Jiangsu Province, China, spanning the years 2019–2024. In both 2019 and 2021, the IRR was 5.95 (95% lower CL = 3.77) for 2019 and 1.96 (95% lower CL = 1.35) for 2021, signifying a significant clustering effect of pinworm infections. Although the estimated IRRs for the years 2020 and 2022 were not significantly greater than 1, corresponding power analyses, conducted under the same one-sided framework with α = 0.05 and an assumed true IRR of 2.0, yielded statistical powers of only 0.08 for 2020 and 0.02 for 2022, not fully ruling out the presence of clustering effect.

### Risk factors for pinworm infection among preschool children in Jiangsu, China

Single factor logistic regression showed that migrant children, age, mother's education level, father's educational level, family income level, drinking unboiled tap water, and bathing frequency were influencing factors for pinworm infection (Table 3). Multiple stepwise logistic regression showed that each one-year increment was associated with a 24.6% reduction in the odds of pinworm infection(OR=0.754, 95%CI = 0.677-0.837). Migrant children exhibited a significantly higher risk of pinworm infection compared to local children (OR=3.911, 95%CI = 2.749-5.610). Preschool children in older age groups were found to have an increased likelihood of pinworm infection (OR=1.443, 95%CI = 1.209-1.730).

### Socioeconomic pathways mediate the association between migrant status and pinworm infection

As shown in Table 4, the total effect of migrant status on pinworm infection was 0.540 (95% CI: 0.420–0.660). The natural indirect effect via parental education and family income level was 0.298 (95% CI: 0.258–0.337), accounting for 55.2% of the total effect, whereas the natural direct effect was 0.242 (95% CI: 0.117–0.368). Parental education and family income level significantly mediated the association. Thus, more than half of the excess infection risk among migrant children is explained by socioeconomic disadvantage.

## Discussion

This study provides a comprehensive analysis of pinworm infections among preschool children in Jiangsu Province from 2019 to 2024, highlighting significant trends and influencing factors. In current study, the overall pinworm infection rate over the six-year period was 0.48%. Significant decreasing trend of preschool pinworm infection was noted in Jiangsu Province, China from 2019 to 2024. There is preschool clustering effect of pinworm infections. There was no significant difference in the pinworm infection rate among symptomatic population and the overall infection rate. Migrant children and

**Table 1. Descriptive statistics of preschool children in Jiangsu Province, China, between 2019 and 2024.**

| | | Total | 2019 | 2020 | 2021 | 2022 | 2023 | 2024 | P a |
|---|---|---|---|---|---|---|---|---|---|
| **Preschool** | Rural | 10305 | 1331 | 933 | 1046 | 2566 | 1833 | 2596 | <0.001 |
| | Urban | 17620 | 2347 | 2296 | 2756 | 3254 | 3940 | 3027 | |
| **Gender** | Male | 14519 | 2000 | 1720 | 1973 | 2921 | 3026 | 2879 | 0.001 |
| | Female | 13406 | 1678 | 1509 | 1829 | 2899 | 2747 | 2744 | |
| **Age** | 2-3 | 4420 | 778 | 439 | 704 | 1309 | 492 | 698 | |
| | 4 | 8017 | 1142 | 864 | 1303 | 1701 | 1446 | 1551 | |
| | 5 | 9269 | 1243 | 1038 | 1300 | 1871 | 1855 | 1962 | |
| | 6-7 | 6219 | 515 | 888 | 485 | 939 | 1980 | 1412 | |
| **Grade** | Lower | 7687 | 1087 | 923 | 948 | 1804 | 1441 | 1484 | <0.001 |
| | Middle | 10219 | 1325 | 1088 | 1687 | 2137 | 1983 | 1999 | |
| | Upper | 10019 | 1266 | 1218 | 1167 | 1879 | 2349 | 2140 | |
| **Father's education level** | Illiterate/Primary school | 807 | 106 | 117 | 60 | 229 | 154 | 141 | |
| | Secondary school | 7012 | 1115 | 1139 | 994 | 1365 | 1287 | 1112 | |
| | Technical secondary school | 7679 | 1241 | 884 | 1300 | 1048 | 1565 | 1641 | |
| | High school/Technical college | 5882 | 617 | 476 | 617 | 1689 | 1254 | 1229 | |
| | Bachelor or higher | 6545 | 599 | 613 | 831 | 1489 | 1513 | 1500 | |
| **Mother's education level** | Illiterate/Primary school | 1003 | 134 | 147 | 101 | 276 | 180 | 165 | |
| | Secondary school | 6703 | 1001 | 1027 | 824 | 1327 | 1371 | 1153 | |
| | Technical secondary school | 4962 | 634 | 463 | 704 | 710 | 1223 | 1228 | |
| | High school/Technical college | 8179 | 1193 | 897 | 1141 | 2013 | 1452 | 1483 | |
| | Bachelor or higher | 7078 | 716 | 695 | 1032 | 1494 | 1547 | 1594 | |
| **Family income level** | Low | 5968 | 785 | 750 | 786 | 1222 | 1326 | 1099 | <0.001 |
| | Medium | 21569 | 2861 | 2449 | 2974 | 4522 | 4385 | 4378 | |
| | High | 388 | 32 | 30 | 42 | 76 | 62 | 146 | |
| **Type of house floor** | Brick/Wood | 25057 | 3092 | 2616 | 3448 | 5194 | 5420 | 5287 | <0.001 |
| | Cement | 2682 | 549 | 571 | 337 | 574 | 333 | 318 | |
| | Soil | 186 | 37 | 42 | 17 | 52 | 20 | 18 | |
| **Wash hand before meals** | Yes | 27512 | 3614 | 3196 | 3765 | 5718 | 5679 | 5540 | 0.005 |
| | No | 413 | 64 | 33 | 37 | 102 | 94 | 83 | |
| **Wash hands after using the toilet** | Yes | 27069 | 3526 | 3146 | 3744 | 5639 | 5582 | 5432 | <0.001 |
| | No | 856 | 152 | 83 | 58 | 181 | 191 | 191 | |
| **Keep fingernails short** | Yes | 27378 | 3608 | 3177 | 3741 | 5686 | 5669 | 5497 | 0.047 |
| | No | 547 | 70 | 52 | 61 | 134 | 104 | 126 | |
| **Suck fingers** | Yes | 5568 | 764 | 568 | 781 | 1116 | 1135 | 1204 | <0.001 |
| | No | 22357 | 2914 | 2661 | 3021 | 4704 | 4638 | 4419 | |
| **Suck toys/pens etc.** | Yes | 3319 | 460 | 314 | 383 | 722 | 715 | 725 | <0.001 |
| | No | 24606 | 3218 | 2915 | 3419 | 5098 | 5058 | 4898 | |
| **Drink unboiled tap water** | Yes | 1773 | 208 | 186 | 186 | 410 | 426 | 357 | <0.001 |
| | No | 26152 | 3470 | 3043 | 3616 | 5410 | 5347 | 5266 | |
| **Have separate towel** | Yes | 26289 | 3470 | 3041 | 3615 | 5445 | 5426 | 5292 | 0.069 |
| | No | 1636 | 208 | 188 | 187 | 375 | 347 | 331 | |
| **Frequency of bathing** | Everyday | 22551 | 2474 | 2656 | 3483 | 4842 | 4314 | 4782 | <0.001 |
| | Every two or three days | 5022 | 1132 | 526 | 301 | 904 | 1372 | 787 | |
| | Once a week or less | 352 | 72 | 27 | 18 | 74 | 87 | 54 | |

*(Continued)*

**Table 1.** (Continued)

|  |  | Total | 2019 | 2020 | 2021 | 2022 | 2023 | 2024 | *P* [a] |
|---|---|---|---|---|---|---|---|---|---|
| **Bed-linen drying-out frequency** | Once a month | 26996 | 3589 | 3153 | 3606 | 5614 | 5587 | 5447 | <0.001 |
|  | Once a season | 804 | 76 | 63 | 172 | 175 | 162 | 156 |  |
|  | Once half year or less | 125 | 13 | 13 | 24 | 31 | 24 | 20 |  |
| **Total** |  | 27925 | 3678 | 3229 | 3802 | 5802 | 5773 | 5623 | / |

a By Chi-square test.

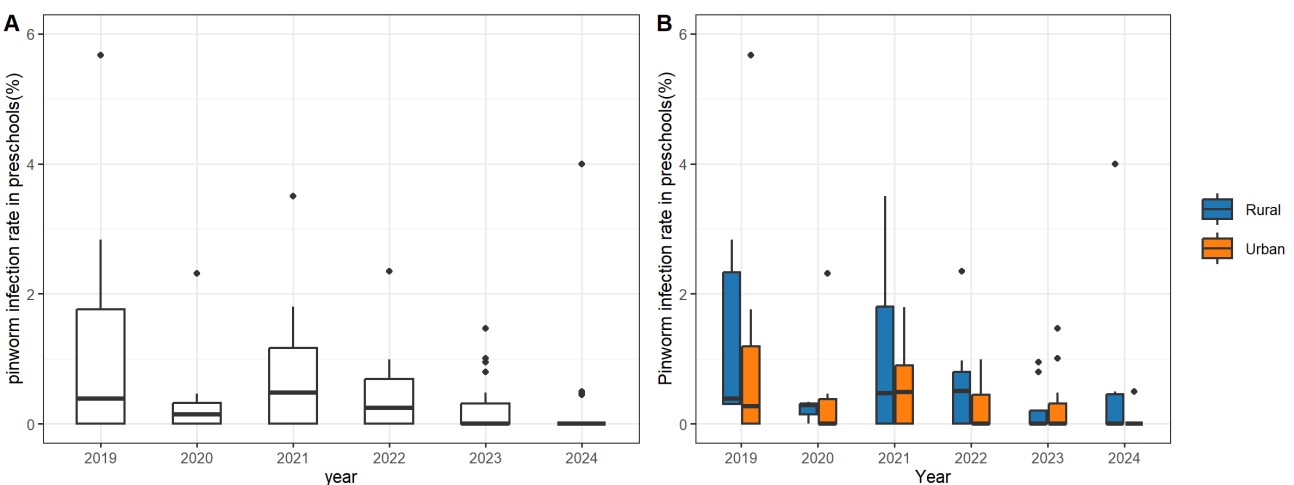

**Fig 2. Pinworm infection rates in preschools in Jiangsu Province, China, between 2019 and 2024.** (A) Observed infection rates in individual preschools. (B) Observed infection rates in individual preschools by urban and rural setting.

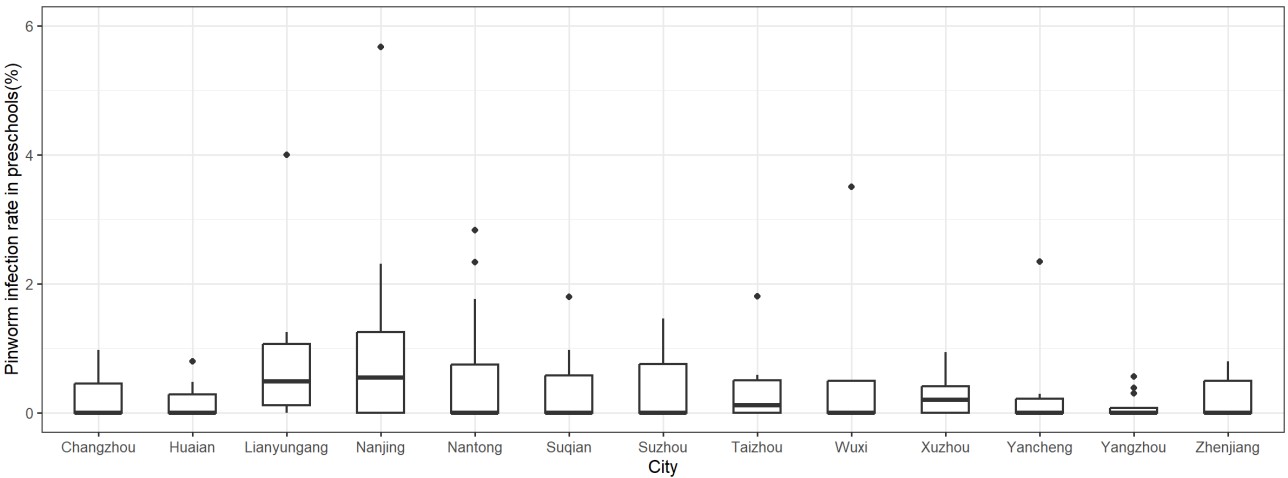

**Fig 3. Pinworm infection rates in preschools in each city of Jiangsu Province, China, between 2019 and 2024.**

**Table 2. The results of permutation approach quantifying the clustering risk of pinworm infections within a preschool, measured as infection rate ratio (IRR) in Jiangsu Province, China, between 2019 and 2022.**

| Year[a] | IRR | 95% lower CL |
|---|---|---|
| 2019 | 5.95 | 3.77 |
| 2020 | 2.27 | 0.85 |
| 2021 | 1.96 | 1.35 |
| 2022 | 2.26 | 0.91 |

[a]The infection rates of 2023 and 2024 were too low to allow for a meaningful calculation of the IRR.

**Table 3. Multivariable stepwise logistic regression of risk factors for pinworm infection in Jiangsu Province, China, between 2019 and 2024.**

| Variable | Single factor logistic regression | | | Multiple stepwise logistic regression[a] | | |
|---|---|---|---|---|---|---|
| | OR | 95%CI | *P* value | OR | 95%CI | *P* value |
| **Year** | 0.709 | 0.639-0.785 | <0.001* | 0.754 | 0.677-0.837 | <0.001* |
| **Preschool type** (Ref = rural preschool) | 0.789 | 0.561-1.117 | 0.176 | | | |
| **Migrant people** (Ref = Local people) | 4.715 | 3.342-6.7007 | <0.001* | 3.911 | 2.749-5.610 | <0.001* |
| **Gender** (Ref = Female) | 1.139 | 0.811-1.608 | 0.453 | | | |
| **Age** | 1.344 | 1.128-1.609 | 0.001* | 1.443 | 1.209-1.730 | <0.001* |
| **Mother's education level** | 0.722 | 0.626-0.830 | <0.001* | | | |
| **Father's education level** | 0.665 | 0.569-0.775 | <0.001* | | | |
| **Family income level** | 0.478 | 0.339-0.679 | <0.001* | | | |
| **Wash hands before meals** | 0.991 | 0.314-6.009 | 0.990 | | | |
| **Wash hands after using the toilet** | 1.383 | 0.522-5.615 | 0.580 | | | |
| **Keep fingernails short** | 0.872 | 0.329-3.543 | 0.815 | | | |
| **Suck fingers** | 0.831 | 0.517-1.278 | 0.421 | | | |
| **Suck toys/pens etc.** | 1.226 | 0.731-1.945 | 0.412 | | | |
| **Drink unboiled tap water** | 1.867 | 1.045-3.095 | 0.023* | | | |
| **Have separate towel** | 0.491 | 0.296-0.878 | 0.010* | | | |
| **Frequency of bathing** | 1.436 | 1.007-1.989 | 0.037* | | | |
| **Bed-linen drying-out frequency** | 1.153 | 0.487-2.156 | 0.702 | | | |
| **Symptom of stomachache** | 0.835 | 0.410-1.512 | 0.585 | | | |
| **Symptom of diarrhea** | 0.829 | 0.254-1.970 | 0.712 | | | |
| **Symptom of fever** | 0.885 | 0.475-1.510 | 0.677 | | | |
| **Symptom of painful urination** | 0.586 | 0.033-2.625 | 0.595 | | | |
| **Symptom of perianal pruritus** | 0.242 | 0.014-1.080 | 0.158 | | | |

[a]AIC = 1587.1

* *P* < 0.05

**Table 4. Mediation effects of parental education and family income level on the association between migrant status and pinworm infection.**

| Effect path | Effect estimate | 95% CI | Proportion mediated (%) |
|---|---|---|---|
| Natural indirect effect (NIE) | 0.298 | 0.258–0.337 | 55.2 |
| Natural direct effect (NDE) | 0.242 | 0.117–0.368 | 44.8 |
| Total effect (TE) | 0.540 | 0.420–0.660 | 100.0 |

older age were found to be risk factors for pinworm infection in preschools. The findings contribute valuable insights into the epidemiology of pinworm infections and the effectiveness of control strategies in preschool settings.

The encouraging downward trend in pinworm infection rates observed over the six-year span aligns with findings from a prior study among Chinese children aged 3–9 years, conducted between 2016 and 2020 [17]. This convergence suggests that control measures may have been effective [18], and underscores the significant role that enhancements in living standards have likely played in this positive development. The spatial pattern analysis using Moran's I did not indicate significant spatial autocorrelation, suggesting that the distribution of infections was likely due to random chance rather than spatial clustering.

The within-preschool clustering risk analysis revealed significant preschool clustering of pinworm infections in both 2019 and 2021. However, no significant clustering was observed in 2020 and 2022. Corresponding power analyses yielded statistical powers of only 0.08 for 2020 and 0.02 for 2022. These values fall far short of the conventional 80% threshold, indicating that the non-significant results in these two years are most likely attributable to inadequate statistical power rather than to a genuine absence of an increased risk. Moreover, the fluctuation in clustering could be influenced by various factors, including seasonal variations, changes in population mobility, or differences in preschool hygiene management and control measures implemented across years. The perception that pinworm infections are more prevalent in gathering settings (such as households and preschools) is widely acknowledged, due to various factors such as limited space, frequent gatherings, and close contact among members. The presence of positive contacts in the same household substantially increased the risk of infection by enteric parasites, supporting infection clustering in the home [19]. However, the phenomenon of infection clustering within preschools has not been substantiated with data. In this study, we introduced the IRR as an indicator to assess whether pinworm infections exhibit clustering within preschool settings. If the IRR is significantly greater than 1, it suggest the presence of a clustering effect within preschools. The within-preschool clustering effect yielded IRRs of 5.95 in 2019 and 1.96 in 2021, indicating that children in preschools were approximately six- and two-fold more likely to acquire pinworm infection, respectively, than those in the wider community. The preschool clustering effect was validated and quantified in this study, revealing the the actual clustering effect present in the real world regarding to pinworm infection.

With economic growth and improved living standards, there has been a cumulative and significant enhancement in residential conditions and hygiene practices, along with an overall increase in intensified disease prevention and control measures and improved sanitation. These factors have contributed to the reduction of pinworm infection and transmission among children over years. The declining trend in pinworm infection rates in preschools over the years aligns with theoretical expectations and are consistent with findings from other studies [17]. Migrant children are also at a higher risk of pinworm infection, which may be closely related to their family conditions, hygiene, and parental education levels [20], highlighting them as a key target group for future pinworm control efforts in preschools. Parental education and family income level significantly mediated the association, while the direct pathway of migrant status remained significant but was attenuated by almost half, indicating that socioeconomic disadvantage partly explains the higher infection risk among migrant children. This pattern aligns with the social determinants of health framework: parental education shapes health-literacy, hygiene behaviour and health-care seeking, whereas income constrains housing quality, overcrowding and access to preventive products. Consistent with previous parasitology studies, children from lower-income families and with unemployed and less educated mothers showed higher risk of intestinal parasitism (OR=6.0, 95%CI = 1.6-22.6; OR=4.5, 95%CI = 2.5-8.2; OR=3.3, 95%CI = 1.5-7.4 respectively) in rural Mexico [21]. Taken together, the observed mediation suggests that socio-economic disadvantage is not a confounder but a core mechanism; consequently, policies that exclusively target deworming campaigns without improving educational and economic opportunities for migrant families are unlikely to eliminate the excess risk. Integrating conditional cash transfers, preschool subsidies and caregiver health-literacy programmes into the current school-based deworming platform could close 50% of the infection gap and produce long-term co-benefits for child development. Additionally, older children have a higher risk of pinworm infection [1], likely associated with their increased activity levels and more exposure to contaminated environments in preschool settings [18,22].

The study provides insights into the epidemiology of pinworm infections in preschool children, which can inform targeted interventions to control and prevent outbreaks. By understanding the trend, clustering, and risk factors, public health policies can be tailored more effectively to mitigate the spread of pinworm infections. This study could also contribute to the broader understanding of enterobiasis and support the development of strategies to protect children's health and well-being in preschool settings.

While this study provides valuable insights, it has some limitations. Firstly, the sensitivity of one single cellophane tape test is around 50%; however, the sensitivity increases to approximately 90% with tests performed on three different mornings [4]. Consequently, one sample per child in this study prrobably underestimate the true infection rate. This underestimation is expected to attenuate absolute prevalence peaks and bias risk-fator estimates toward the null. Because exposure groups are nondifferentially misclassified, diluting observed associations with hygiene-related variables. Thus, we re-checked all positive and 10% of negative samples to ensure the detection rate. Resource constraints precluded repeated sampling in the present survey; future studies should collect at least three specimens per child to minimise false-negative results and to quantify infection intensity accurately. Secondly, non-random sampling of preschools may have diminished the sample's representativeness of all preschools across Jiangsu Province, potentially obscuring a comprehensive view of the real world. Implementing random sampling of preschool is anticipated to bolster the credibility of future investigations into pinworm infections.

## Conclusion

Pinworm prevalence has declined over time; nevertheless, pinworm infections among preschool children demonstrate a within-preschool clustering effect. Multivariate analyses identified migrant children and older age as independent risk factors. This study help understand the epidemiological trend, within-preschool clustering pattern, and risk factors of pinworm infection in early childhood, thereby informing targeted, precision interventions to safeguard paediatric health. The findings show that even in a low-prevalence region, pinworm can hide in socially disadvantaged pockets; targeting these specific populations with hygiene upgrades and periodic screening could accelerate local elimination and serve as a model for other regions striving to eliminate neglected helminth infections.

## Supporting information

**S1 Fig. Spatial autocorrelation (global Moran's I) report (the Basemap layer/shapefiles can be accessed at https://gadm.org).**
(TIF)

**S1 Table. STROBE statement—Checklist of items that should be included in reports of *cross-sectional studies.***
The filled checklist is based on the STROBE Statement-Checklist of items that should be included in reports of observational studies, developed by the STROBE Initiative, https://www.strobe-statement.org/.
(DOCX)

## Acknowledgments

We would like to thank all centers for disease control and prevention of study counties for their involvement in this study.

## Author contributions

**Conceptualization:** Fanzhen Mao, Xuecheng Li, Yaobao Liu, You Li, Jun Cao.

**Data curation:** Fanzhen Mao, Xuecheng Li, Yaobao Liu, You Li.

**Formal analysis:** Fanzhen Mao, Xuecheng Li, Sheng Ye.

**Funding acquisition:** Fanzhen Mao, Jun Cao.

**Investigation:** Fanzhen Mao, Xuecheng Li, Feng Tang, Bixian Ni, Qiang Zhang, Jiayao Zhang.

**Methodology:** Fanzhen Mao, Yaobao Liu, Jun Cao.

**Project administration:** Yaobao Liu, Jun Cao.

**Resources:** Yaobao Liu, You Li, Jun Cao.

**Software:** Fanzhen Mao, Xuecheng Li, Sheng Ye.

**Supervision:** Yaobao Liu, You Li, Jun Cao.

**Validation:** Yaobao Liu, You Li, Jun Cao.

**Visualization:** Fanzhen Mao, Xuecheng Li, Sheng Ye.

**Writing – original draft:** Fanzhen Mao, Xuecheng Li.

**Writing – review & editing:** Fanzhen Mao, Yaobao Liu, You Li, Jun Cao.

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
