## [Decision Letter · Decision Letter 0]

12 Nov 2025

* A rebuttal letter that responds to each point raised by the editor and reviewer(s). You should upload this letter as a separate file labeled 'Response to Reviewers '. This file does not need to include responses to any formatting updates and technical items listed in the 'Journal Requirements' section below.

* A marked-up copy of your manuscript that highlights changes made to the original version. You should upload this as a separate file labeled 'Revised Manuscript with Track Changes '.

Manuscript

Shaden Kamhawi

co-Editor-in-Chief

Paul Brindley

co-Editor-in-Chief

**Journal Requirements:**

At this stage, the following Authors/Authors require contributions: Fanzhen Mao, Xuecheng Li, Sheng Ye, Feng Tang, Bixian Ni, Qiang Zhang, Jiayao Zhang, Yaobao Liu, You Li, and Jun Cao. Please ensure that the full contributions of each author are acknowledged in the "Add/Edit/Remove Authors" section of our submission form.

3) Some material included in your submission may be copyrighted. According to PLOSu2019s copyright policy, authors who use figures or other material (e.g., graphics, clipart, maps) from another author or copyright holder must demonstrate or obtain permission to publish this material under the Creative Commons Attribution 4.0 International (CC BY 4.0) License used by PLOS journals. Please closely review the details of PLOSu2019s copyright requirements here: PLOS Licenses and Copyright. If you need to request permissions from a copyright holder, you may use PLOS's Copyright Content Permission form.

Potential Copyright Issues:

i) Figures S1, and S2. Please (a) provide a direct link to the base layer of the map (i.e., the country or region border shape) and ensure this is also included in the figure legend; and (b) provide a link to the terms of use / license information for the base layer image or shapefile. We cannot publish proprietary or copyrighted maps (e.g. Google Maps, Mapquest) and the terms of use for your map base layer must be compatible with our CC BY 4.0 license.

4) Please amend your detailed Financial Disclosure statement. This is published with the article. It must therefore be completed in full sentences and contain the exact wording you wish to be published.

1) State the initials, alongside each funding source, of each author to receive each grant. For example: "This work was supported by the National Institutes of Health (####### to AM; ###### to CJ) and the National Science Foundation (###### to AM).".

**Reviewers' comments:**

Reviewer's Responses to Questions

**Key Review Criteria Required for Acceptance?**

**Methods**

-Are the objectives of the study clearly articulated with a clear testable hypothesis stated?

-Is the study design appropriate to address the stated objectives?

-Is the population clearly described and appropriate for the hypothesis being tested?

-Is the sample size sufficient to ensure adequate power to address the hypothesis being tested?

-Were correct statistical analysis used to support conclusions?

-Are there concerns about ethical or regulatory requirements being met?

Reviewer #1: (No Response)

Reviewer #2: The objectives are well articulated and the hypothesis was stated (Line 161). The study design was appropriate within the limits specified by the authors. The description of the population is adequate. The sample size was appropriately estimated. The statistical analysis are adequate, but a suggestion on improvement is provided. No concern about ethical defects.

**Results**

-Does the analysis presented match the analysis plan?

-Are the results clearly and completely presented?

-Are the figures (Tables, Images) of sufficient quality for clarity?

Reviewer #1: (No Response)

Reviewer #2: The analysis match the analysis plan. The result are clearly presented. The Figures and tables are sufficient but some suggestions for modification are suggested

**Conclusions**

-Are the conclusions supported by the data presented?

-Are the limitations of analysis clearly described?

-Do the authors discuss how these data can be helpful to advance our understanding of the topic under study?

-Is public health relevance addressed?

Reviewer #1: (No Response)

Reviewer #2: The conclusion is supported by the data. Limitations were specified. The discussion could be improved. Particularly, the authors should compare with the outcomes of similar studies, especially on the risk factors.

**Editorial and Data Presentation Modifications?**

Reviewer #1: (No Response)

Reviewer #2: Title: “between 2109-2024” instead of “in 2019-2024”

The map in figure S1 can be moved to the main text because it is very useful.

E. vermicularis is not an STH. Please correct this in the abstract.

Is E. vermicularis infection also the most prevalent? I doubt. Please support this claim by citing a reliable literature.

Line 120-122: Instead of spring and autumn, specify the months

Line 147 and elsewhere: Define “raw water”. Is this tap water?

Line 188: correct to “Descriptive statistics are presented”. Correct throughout the results section

Please specify the guidelines used to set the cut-off for income to low medium and high

Line 204: Figure title should read “Pinworm infection rates in preschools in Jiangsu Province, China, between 2019-2024.” Correct other figure titles accordingly

Figure 1 can be presented using a box plot. The information of both A and B will fit nicely in one box plot (with even more properties of the data, such as median included). This will also allow for the presentation of the rates for in urban vs rural (C). Apply same logic for Figure 2

Line 288-290: Based on this logic, the variance inflation factor (VIF) for the logistic regression should be reported. The increased risk in infection among migrant children may have been caused by other factors, which could inflate the variance. The VIF will tell whether one factor is affecting another.

Please include title page in the supplementary materials

**Summary and General Comments**

Reviewer #1: In this manuscript, Mao et al., adopts a 6-year (2019–2024) repeated cross-sectional multi-center design conducted across 45 counties and 104 preschools with a total sample size of 27,925 children in Jiangsu Province, China. This long-term, large-scale framework addresses the scarcity of extended temporal data on pinworm (Enterobius vermicularis) infection in preschool settings. The study also systematically differentiates between rural and urban preschools in sampling, which is also valuable for tailoring location-specific interventions. However, there are still some limitations that can be improved in the future revision.

1) The study collects only one adhesive cellophane tape swab per child, despite noting that single-test sensitivity is ~50% (vs. ~90% for three consecutive daily tests). This likely leads to underestimation of the true infection rate, which could skew trend analyses (e.g., masking subtle fluctuations) or risk factor associations (e.g., diluting the strength of hygiene-related correlations). The authors should discuss this underestimation of the true infection rate.

2) While the study identifies migrant status and age as independent risk factors, it provides limited mechanistic explanation for why migrant children are at higher risk. For example, it mentions "family conditions and parental education" but does not analyze whether these factors (e.g., lower parental education, poorer living conditions) mediate the association between migrant status and infection. Could the authors conduct a mediation analysis to assess whether factors like parental education or family income explain the migrant status-infection association?

Reviewer #2: The manuscript is well written and the data presented is beneficial. The statistics and design of the work are the main strengths of the manuscript. The discussion is a mild weakness that should be easily improved.

PLOS authors have the option to publish the peer review history of their article (what does this mean? ). If published, this will include your full peer review and any attached files.

**Do you want your identity to be public for this peer review?** For information about this choice, including consent withdrawal, please see our Privacy Policy .

Reviewer #1: No

Reviewer #2: No

**Figure resubmission:**

**Reproducibility:** To enhance the reproducibility of your results, we recommend that authors of applicable studies deposit laboratory protocols in protocols.io, where a protocol can be assigned its own identifier (DOI) such that it can be cited independently in the future. Additionally, PLOS ONE offers an option to publish peer-reviewed clinical study protocols. Read more information on sharing protocols at https://plos.org/protocols?utm_medium=editorial-email&utm_source=authorletters&utm_campaign=protocols

---

## [Editor Report · Decision Letter 1]

26 Nov 2025

Dear Mr. Liu,

We are pleased to inform you that your manuscript 'Understanding the trends, clustering, and risk factors of pinworm infection in preschool settings: a repeated cross-sectional multi-center study between 2019 and 2024' has been provisionally accepted for publication in PLOS Neglected Tropical Diseases.

Best regards,

María Victoria Periago

Section Editor

jong-Yil Chai

Section Editor

Shaden Kamhawi

co-Editor-in-Chief

Paul Brindley

co-Editor-in-Chief

---

## [Editor Report · Acceptance letter]

Dear Mr. Liu,

We are delighted to inform you that your manuscript, "Understanding the trends, clustering, and risk factors of pinworm infection in preschool settings: a repeated cross-sectional multi-center study between 2019 and 2024," has been formally accepted for publication in PLOS Neglected Tropical Diseases.

Best regards,

Shaden Kamhawi

co-Editor-in-Chief

Paul Brindley

co-Editor-in-Chief
